# INDIANROAD: A VIDEO DATASET OF DIVERSE ATOMIC VISUAL ELEMENTS IN DENSE AND UNPREDICTABLE ENVIRONMENTS

## ABSTRACT

Most existing traffic video datasets including Waymo Sun et al. (2020) are structured, focusing predominantly on Western traffic, which hinders global applicability. Specifically, most Asian scenarios are far more complex, involving numerous objects with distinct motions and behaviors. Addressing this gap, we present a new dataset, IndianRoad, designed for evaluating perception methods with high representation of Vulnerable Road Users (VRUs: e.g. pedestrians, animals, motorbikes, and bicycles) in complex and unpredictable environments. IndianRoad is a manually annotated dataset encompassing 16 diverse actor categories (spanning animals, humans, vehicles, etc.) and 16 action types (complex and rare cases like cut-ins, zigzag movement, U-turn, etc.), which require high reasoning ability. IndianRoad densely annotates over **13 million** bounding boxes (bboxes) actors with identification, and more than 1.6 million boxes are annotated with both actor identification and action/behavior details. The videos within IndianRoad are collected based on a broad spectrum of factors, such as weather conditions, the time of day, road scenarios, and traffic density. IndianRoad can benchmark video tasks like Tracking, Detection, Spatiotemporal Action Localization, Language-Visual Moment retrieval, and Multi-label Video Action Recognition. Given the critical importance of accurately identifying VRUs to prevent accidents and ensure road safety, in IndianRoad, vulnerable road users constitute **41.13%** of instances, compared to 23.71% in Waymo Sun et al. (2020). IndianRoad provides an invaluable resource for the development of more sensitive and accurate visual perception algorithms in the complex real world. Our experiments show that existing methods suffer degradation in performance when evaluated on IndianRoad, highlighting its benefit for future video recognition research.

## 1 INTRODUCTION

Video recognition research has made significant progress in recent years, enabling applications such as autonomous driving, surveillance systems, and human-computer interaction. At the core of these advancements lies the development of comprehensive and challenging datasets that facilitate the training, evaluation, and benchmarking of novel algorithms. However, the focus has predominantly been on structured environments, featuring human-centric activities Liu et al. (2023) and relatively simplistic scenes that, while beneficial, do not encapsulate the breadth of complexities inherent in natural environments Gu et al. (2018); Kay et al. (2017). This dissimilarity between existing training datasets and the real-world distribution hinders the generalization capabilities of video recognition models, ultimately limiting their effectiveness when applied to multifaceted and unpredictable real-world situations.

In terms of most existing datasets on video recognition research, there are some limitations:

- *Limited Scope*: Most existing datasets primarily focus on human actors performing isolated actions (one action in one clip) in simplistic and controlled settings. This narrow scope restricts the ability of models to generalize to diverse scenarios with varying object categories, environmental factors, and complex interactions.

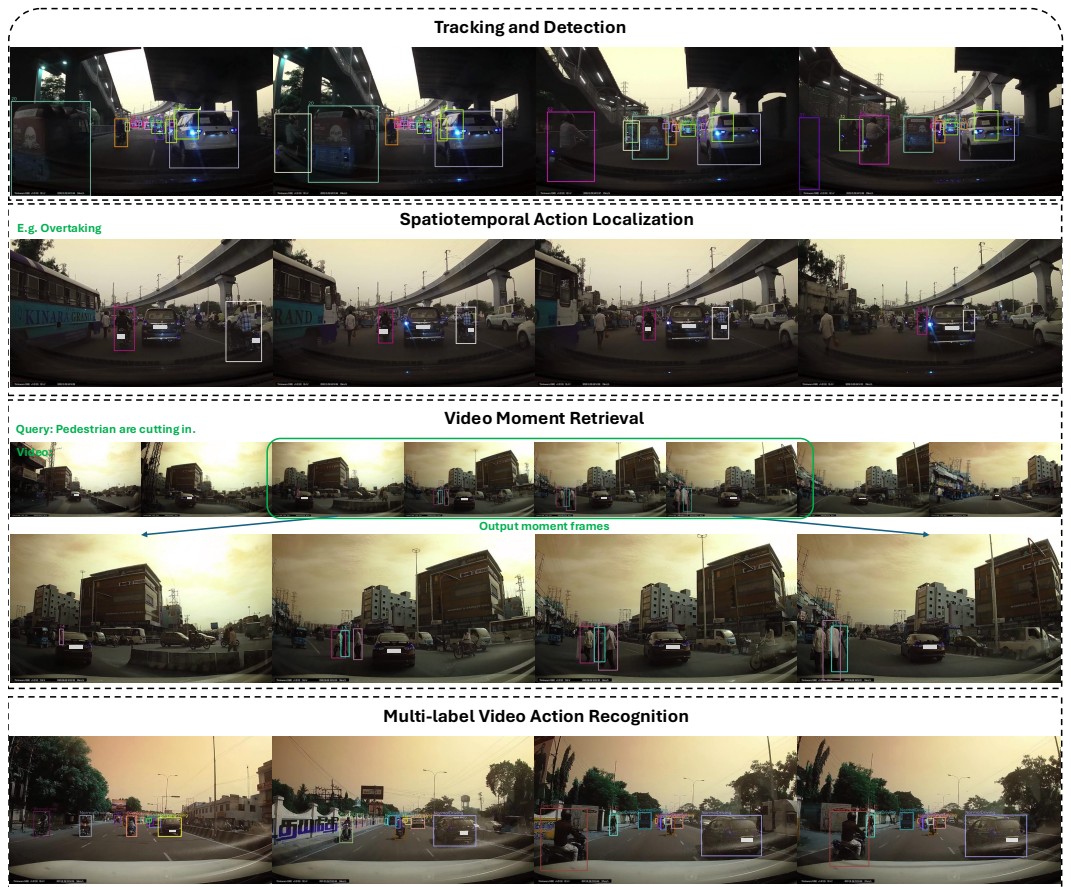

Figure 1: **Tasks Overview.** We use IndianRoad for various video recognition tasks, including Tracking, Detection, Video Moment Retrieval, Spatiotemporal Action Localization, and Multi-label Video Action Recognition. Our large-scale dataset is made up of complex environments that are densely annotated. Each bounding box (bbox) corresponds to an actor, and the text above each bbox serves as either the tracking ID or indicates the associated action.

- *Lack of Unstructured Environments*: Some datasets , while encompassing a broader range of activities, predominantly feature structured settings with clear foreground-background separation. This lack of real-world complexity, such as cluttered scenes, occlusions, and dynamic lighting, hinders the development of robust perception models.

- *Sparse Annotations*: Many datasets lack fine-grained information about object locations, interactions, and temporal relationships. This hinders the evaluation of various tasks like Spatiotemporal Action Localization and Video Moment Retrieval, which require detailed temporal and spatial annotations.

In human-centric datasets, AVA Gu et al. (2018) has atomic visual human actions that are localized in space and time, including interactions with people and objects. The mutual interactions and relationships in this dataset make AVA a hard dataset for Spatiotemporal Action Localization even nowadays. AVA has been collected from movies in structured scenes and the human-centric action is relatively simple (e.g. stand, watch, sit, walk). Inspired by AVA, we want to build an ego-car-centric dataset and annotate the surrounding agents' actions in space and time dimensions. Furthermore, we chose India to collect the metadata to ensure the precious density and actor diversity, which also allows the high representation of VRUs.

Therefore, we introduce a new dataset, IndianRoad, where every visible object is annotated and considered an atomic visual element. It is specifically designed to evaluate perception methods in unstructured environments that are more indicative of real-world scenarios. The unstructured en-

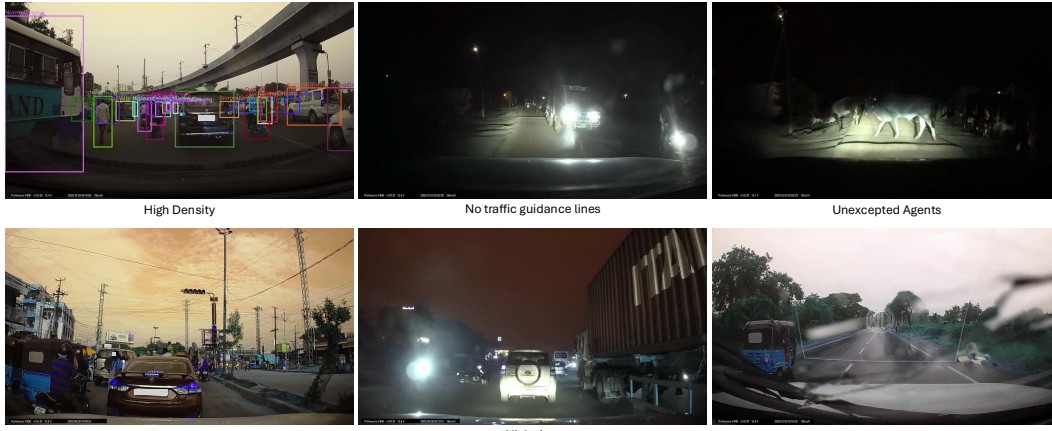

Figure 2: **Challenging Characteristics of IndianRoad:** These videos correspond to different times of the day with different brightness, different geographical landforms from city and rural areas, high density and unpredictable road conditions, diverse actors including humans, animals, vehicles, etc.

vironments in IndianRoad cover different geographical landforms, diverse actors (not only humans but also animals, vehicles, etc.), and complex actions (cut-in, overtaking, u-turn, etc.). As shown in Fig. 2, IndianRoad prioritizes replicating the richness and complexities encountered in real-world situations. We highlight its applicability to various video recognition tasks as shown in Fig. 1, including Tracking, Detection, Video Moment Retrieval, Spatiotemporal Action Localization, and Multi-label Video Action Recognition. In each case, IndianRoad has its distinctive features and novel challenges. Some key characteristics of IndianRoad include:

- *Less predictable and Dense Environments*: IndianRoad features videos captured in diverse real-world settings, encompassing various weather conditions, times of day, road scenarios, and traffic densities. This inherent complexity better reflects the challenges encountered in practical applications.

- *Rich Annotations*: IndianRoad provides dense annotations, including over 13 million bounding boxes (bboxes) for actors and over 1.6 million bboxes encompassing both actor and action details (Table. 1). We also offer actors' GPS information and the keyframe for the action. This comprehensive annotation allows for the evaluation of a wider range of potential tasks.

- *Diverse Actor Categories*: IndianRoad extends beyond human-centric datasets, incorporating 16 diverse actor categories. This diversity fosters the development of models capable of generalizing beyond a limited set of actor types.

- *Complex Actions*: Compared with human-centric simple actions (e.g stand, watch, sit, walk), IndianRoad has more complex actions (e.g. cut-in, overtaking, u-turn, ZigzagMovement), which require higher reasoning ability for perception models.

- *Vulnerable Road Users (VRUS)*: IndianRoad has a higher representation of vulnerable road users (VRUs), constituting 41.13% compared to 23.71% in Waymo Sun et al. (2020). This is a precious property to prevent accidents and ensure road safety.

Table 1: IndianRoad Characteristics: We annotate 16 types of actions performed by 16 types of actors. We highlight the maximum and average number of actions and actors per frame. LaneChanging(m) denotes lane changing on roads with clear lane markings.

| Property | Values |
|---|---|
| Basic Information | Location: India (urban and semi-urban settings) |
| Action Types (16) | NormalDriving, Yield, Cutting, LaneChanging(m), OverSpeeding, WrongTurn, TrafficLight, WrongLane, ZigzagMovement, LaneChanging, OverTaking, Keep, LeftTurn, RightTurn, UTurn, Breaking |
| Action Statistics | Max action num per frame: 40, Average action num per frame: 6.7
Max unique action num per frame: 6, Average unique action num per frame: 2.0 |
| Types of Actors (16) | AgricultureVehicle, Animal, Bicycle, Bus, Car, ConstructionVehicle, EgoVehicle, MotorBike, MotorizedTricycle, MultiWheeler, Pedestrian, Scooter, Tractor, TriCycle, Truck, Van |
| Actor Statistics | Max actor num per frame: 40, Average actor num per frame: 6.5
Max unique actor num per frame: 10, Average unique actor num per frame: 3.9 |

We highlight the advantages of IndianRoad for five video tasks:

**Tracking:** Compared to datasets like MOT17 Milan et al. (2016), which primarily focus on tracking pedestrians and vehicles in controlled settings, IndianRoad's diverse actors occur under a variety of illumination conditions and provide a more significant challenge for tracking algorithms. This allows for the evaluation of robust tracking methods capable of handling occlusions, cluttered scenes, and dynamic environments. From our experiments, ARTrack Wei et al. (2023) performs 23.7% worse on IndianRoad than GOT-10k, which highlights the complexity of IndianRoad as compared to other datasets.

**Detection:** Datasets like COCO Lin et al. (2014b) and Pascal VOC Everingham et al. (2010) have been instrumental in advancing object detection methods. While these datasets include a variety of object categories, they often lack the contextual complexity and scene diversity found in IndianRoad (e.g. intricate street-scapes at different times of day, higher representation of VRUs, such as pedestrians, animals, motorbikes, and bicycles, compared to vehicles). With its extensive annotations encompassing over 13 million bounding boxes, IndianRoad offers a unique challenge to detection algorithms, pushing the boundaries of what these models can recognize and how well they can adapt to diverse and unstructured environments. In our experiments, Swin-T Liu et al. (2021) outperforms by 18% on the COCO dataset, as compared to IndianRoad. This highlights the complexity of IndianRoad.

**Spatiotemporal Action Localization (STAL):** Spatiotemporal action localization requires algorithms to not only recognize specific actions but also pinpoint their occurrence within both the spatial and temporal domains of video content. Datasets like AVA Gu et al. (2018) have laid the groundwork for this task. It is, however, a movie-human-centric dataset, meaning the video clips in AVA are sourced from movies, which might not perfectly reflect the full diversity of real-world scenarios. This could potentially limit the generalizability of models trained on this dataset. In contrast, IndianRoad introduces a richer layer of complexity by featuring the actions performed by different actor categories in unstructured settings. This complexity is important for developing models that can understand and interpret actions in a manner that is similar to human perception. In our experiments, ACAR-Net Pan et al. (2021) gets 6.3% mAP accuracy on IndianRoad versus 33.3% on AVA v2.2, which highlights the challenging scenarios in IndianRoad.

**Video Moment Retrieval (VMR):** Moment retrieval involves identifying specific moments within a video that correspond to given queries, often described in natural language. While datasets such as DiDeMo Hendricks et al. (2017) are widely used for this task, IndianRoad consists of videos of more complicated and cluttered environments. These scenarios not only demand accurate video understanding but also necessitates sophisticated language processing capabilities to interpret the queries and localize the relevant moments within real-world video content. In our experiments, CG-DETR Moon et al. (2023) obtains 5.1 R1@0.5 on IndianRoad (versus 58.4 on Charades-STA). This implies that video moment retrieval is still a challenging problem in the unstructured environment.

**Multi-label Video Action Recognition (M-VAR):** Multi-label video action recognition is a task that demands the identification of multiple actions within a single video clip. Existing datasets like Charades Sigurdsson et al. (2016b) have been widely used for this video task. IndianRoad's video segments with multiple actions occurring within the densely populated and unstructured scenes offer a challenging testbed for algorithms. In our experiments, SlowFast Feichtenhofer et al. (2019) gets 41.0 mAP accuracy on IndianRoad, while achieving 4.2% higher performance on Charedes.

Overall, IndianRoad offers a valuable resource for researchers aiming to develop robust and generalizable video recognition models that can work well in real-world scenarios. IndianRoad's rich annotations make it suitable for evaluating various video recognition tasks. Check the appendix for more related works.

## 2 INDIANROAD DATASET

### 2.1 DATA COLLECTION

To meet the requirement, data collection was meticulously executed within a defined geographic perimeter encompassing the urban and suburban zones of India. The selection of numerous suburban locations was strategic, aiming to encompass a broad spectrum of road environments, including

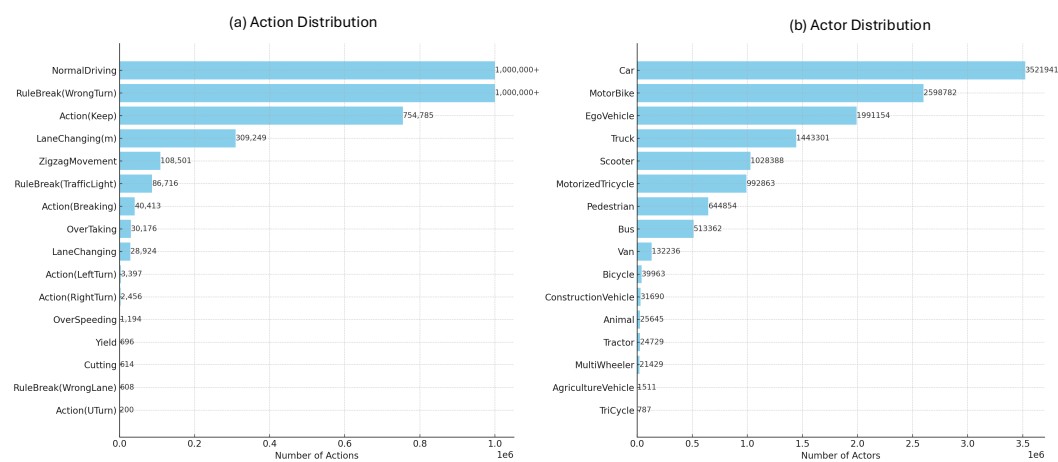

Figure 3: **Annotation Statistic.** The actor and action distribution for IndianRoad, includes a wide-ranging and rich taxonomy of 16 agents and 16 action categories. This dual focus on both the breadth of agent and action types and the depth of instances allows for more robust and effective training of video recognition models.

both rural pathways and those lacking structured design or layout. To capture this data, our equipment consisted of two wide-angle Thinkware F800 dashcams. These devices were installed on two vehicles, specifically an MG Hector and a Maruti Ciaz, chosen for their operational reliability in diverse road conditions. The dashcams are equipped with sensors boasting a resolution of 2.3 megapixels, alongside a comprehensive 140-degree field of view, ensuring wide coverage of the surrounding environment. Video capture was conducted at a high-definition quality, with a resolution of 1920x1080 pixels, and a smooth playback of 30 frames per second was maintained to accurately document the dynamic road conditions.

An integral component of our capture system was the dashcam's embedded positioning technology, which provided precise GPS coordinates. This functionality was essential for the transformation of these coordinates into world frame references, facilitating a coherent geographical mapping of the data collected. Additionally, the system's synchronization capability ensured seamless integration of video and GPS data, enhancing the reliability of the spatial information.

The resultant dataset comprises 1231 video clips, each spanning one minute in duration. These clips are accompanied by corresponding information such as the behaviors observed, the type of road, and the overall scene structure. For granular details at the frame level, we offer bounding boxes, precise GPS coordinates, and the behaviors of moving agents within the frame.

IndianRoad is methodically organized to support efficient querying, facilitated by a range of filters. Users can refine searches based on criteria such as road type, traffic density, geographic area, prevailing weather conditions, and observed behaviors.

## 2.2 ANNOTATIONS

In our research, we undertook a meticulous process of manually annotating video data using the Computer Vision Annotation Tool (CVAT) CVAT.ai Corporation (2023), a widely recognized tool for video and image annotation in the field of computer vision. Our annotation process was comprehensive, covering a broad spectrum of labels that are crucial for the development and evaluation of autonomous driving systems. These labels include:

- Bounding Boxes: For each agent visible in the video footage, we provided bounding boxes. These are essential for object detection tasks, enabling algorithms to identify and track the location and dimensions of various agents within the scene.

- Actions and Maneuvers: The dataset catalogues specific vehicle actions and maneuvers, including left/right turns, U-turns, overtaking, braking, etc. This is critical for predicting vehicle behavior and for training systems in decision-making.

- Actor Class IDs: We classified each agent into distinct categories, assigning a unique class ID to facilitate the differentiation and identification of various types of agents, such as vehicles, pedestrians, and bicycles.

- Rare and Interesting Behaviors: We have specifically noted instances of rare and unusual behaviors among traffic participants. Capturing these scenarios is important for preparing autonomous systems to handle edge cases safely.

- GPS Trajectories for the Ego-Vehicle: The dataset includes precise GPS trajectories for the ego-vehicle, providing valuable data on its movement and position over time.

- Environmental Conditions: Annotations in this category encompass weather conditions, time of day, traffic density, and the diversity of traffic participants. This information is crucial for testing and developing autonomous systems that can operate under a wide range of environmental scenarios.

- Road Conditions: We have annotated various aspects of road conditions including whether the environment is urban or rural, the presence and visibility of lane markings, and more. This aids in assessing how different road conditions affect the performance of autonomous driving technologies.

- Road Network Features: Detailed annotations of road network features such as intersections, roundabouts, and traffic signals are included. These are vital for navigation algorithms and for understanding traffic flow and driving behaviors in complex road networks.

- Camera Intrinsic Matrix: For depth estimation and generating accurate trajectories of surrounding vehicles, we include the camera intrinsic matrix. This technical detail enables the conversion of 2D images into 3D representations, essential for spatial understanding and accurate positioning of objects in relation to the ego-vehicle.

As shown in Fig. 3, our dataset stands out with its wide-ranging and rich taxonomy of agent and action categories. This diversity is crucial for ensuring perception systems can operate safely and efficiently in varied and unpredictable environments. Furthermore, our dataset is meticulously designed to capture a wide variety of action categories and a high number of instances within each category. This dual focus on the breadth of agents, action types, and depth of instances allows for more robust and effective training of video recognition models.

Following the popular dataset Waymo Sun et al. (2020), we obey the widely used data collection and use similar rules. We collected this data for Non-commercial Purposes including the use of the Dataset to perform benchmarking for purposes of academic or applied research publication. To protect privacy, we will hide identities by blurring the faces of persons and license plates of vehicles in the dataset with blurring techniques (face detection method Retinaface Deng et al. (2020), license plates method Yan et al. (2023)) to ensure that the identity of pedestrians and other individuals (cars) is not discernible.

## 3 DATASETS FOR DIFFERENT TASKS AND EXPERIMENTS

### 3.1 TRACKING

**Dataset Structure:** IndianRoad contains annotations for multiple objects, so we can construct sequences of frames in which the same object is present. Of IndianRoad's 1231 videos, we can construct 44.8k frame sequences suitable for tracking.

**Experiment Setting:** To assess visual object tracking on IndianRoad, we use Autoregressive Visual Tracking (ARTrack) Wei et al. (2023), which boasts SOTA performace on GOT-10k Huang et al. (2019), TrackingNet Muller et al. (2018), LaSOT Zhan et al. (2019), and LaSOT$_{ext}$ Fan et al. (2021). We utilize a publicly released "ARTrack-256" checkpoint, pretrained on COCO Lin et al. (2014a), GOT-10k, LaSOT, and TrackingNet. ARTrack handles single object tracking as a coordinate sequence interpretation task using a template region from an initial frame. ARTrack does not

determine when a tracking ID is visible, so we only use sequences of frames in which the same object is present. From the 231 videos in the IndianRoad validation split, we filter 5227 frame sequences in which one tracking ID is continuously present for at least 60 frames. This filtering of sequences gives ARTrack a slight advantage because it is a harder task to both detect visibility and track over time. Bounding box predictions from ARTrack-256 are compared to ground truth using average area overlap (AO), success rate at 0.5 IoU ($SR_{0.5}$), and success rate at 0.75 IoU ($SR_{0.75}$).

**Results:** We find that IndianRoad is comparable to GOT-10k in AO but more challenging for both success rate metrics. For $SR_{0.75}$, ARTrack performs 23.7% worse on IndianRoad than GOT-10k, despite our preprocessing to keep the same object present in each frame sequence. While ARTrack performs well on the AO metric, the degradation in SR implies that the tracker may generate bboxes larger than the actual object or that it has increased sensitivity to object appearance changes. For example, illumination variations or pose changes can cause inaccurate predictions in some frames even when average overlap remains decent. We believe IndianRoad becomes even more challenging when one considers the entire video sequence, requiring the tracking of multiple objects as they move in and out of the frame.

Table 2: Comparison of Various Tracking Datasets. IndianRoad is comparable to GOT-10k in AO but more challenging for both success rate metrics. For $SR_{0.75}$, ARTrack performs 23.7% worse on IndianRoad than GOT-10k, despite our preprocessing to keep the same object present in each frame sequence.

| Dataset | Sequence number | Annotation | SOTA Performance | | |
|---|---|---|---|---|---|
| MOT17 Sun et al. (2019) | 14 | Manual | 65.8@HOTA | 81.0@MOTA | 81.1@IDF1 |
| TAO Dave et al. (2020) | 2.9k | Manual | 47.2@TETA | 66.2@LocA | 46.2@AssocA |
| LaSOT Zhan et al. (2019) | 1.4k | Manual | 74.0@AUC | 82.8@PNor | 81.1@P |
| TrackingNet Muller et al. (2018) | 30k | Semi-auto | 86.1@AUC | 90.4@PNor | 86.2@P |
| GOT-10k Huang et al. (2019) | 10k | Manual | 79.5@AO | 87.8@SR50 | 79.6@SR75 |
| IndianRoad | 44.8k | Manual | 72.6@AO | 70.2@SR50 | 47.2@SR75 |

## 3.2 DETECTION

**Dataset Structure:** For detection, we have 13 million annotated bounding boxes with identifying actors in 16 categories. We prepare them in COCO format.

**Experiment Setting:** For the object detection step, we use the Swin-T detector, generated by combining a Cascade R-CNN Cai & Vasconcelos (2018) with a Swin-T Liu et al. (2021) backbone. The model is pre-trained on ImageNet and MS COCO, and fine-tuned on IndianRoad using the same settings as Swin-T Liu et al. (2021): multi-scale training Carion et al. (2020) (resizing the input with the shorter side between 480 and 800 and the longer side at most 1333), AdamW optimizer (initial learning rate of $1e-4$, weight decay of 0.05, and batch size of 16), and $1\times$ schedule (12 epochs).

**Results:** In this paper, our objective is not to enhance object detection within the IndianRoad dataset. Instead, we aim to demonstrate the decline in perception performance in unstructured situations. Delving into the reasons behind this performance drop and identifying methods to better object detection in these chaotic environments is not covered in our current research community. The results show that our IndianRoad dataset is more challenging than the existing datasets.

Table 3: Comparison of Various Detection Datasets. Compared with COCO, with the same setting, Swin-T performs 18% better on the COCO Dataset. The results show that our IndianRoad dataset is more challenging than the existing datasets.

| Dataset | Bbox # | Size | Frame # | Annotation | Weather | Country | SOTA (mAP) |
|---|---|---|---|---|---|---|---|
| COCO Lin et al. (2014a) | 2.5M | Variable | 330K images | Manual | Various | / | 66.0 |
| Pascal VOC Everingham et al. (2010) | 20K | Variable | 11K images | Manual | / | / | 89.3 |
| Waymo Sun et al. (2020) | 11M | Variable | / | Manual/Auto | Various | USA | 41.6 |
| COCO-Swin-T Lin et al. (2014a) | 2.5M | Variable | 330K images | Manual | Various | / | 50.5 |
| IndianRoad | 13M | 1920x1280 | 2M images | Manual | Has Bad weather | India | 32.5 |

Table 4: Statistics of datasets for Video Moment Retrieval task. The CG-DETR method only gets 5.1 R1@0.5 on IndianRoad (58.4 on Charades-STA), and the perception performance degrades significantly illustrating that Video Moment Retrieval is still a challenging problem in the unstructured environment.

| Dataset | #Videos | #Queries | Duration | Domain | Source | R1@0.5 |
|---|---|---|---|---|---|---|
| DiDeMo Hendricks et al. (2017) | 10,464 | 40,543 | 30s | Open | Flickr | 33.4 |
| Charades-STA Sigurdsson et al. (2016a) | 9,848 | 16,128 | 31s | Daily activities | Homes | 60.8 |
| TACOS Regneri et al. (2013) | 127 | 18,818 | 296s | Cooking | Lab Kitchen | 41.54 |
| ActivityNet-Captions Wang et al. (2018) | 19,209 | 71,957 | 180s | Open | YouTube | 60.57 |
| Charades-STA (CG-DETR) Sigurdsson et al. (2016a) | 9,848 | 16,128 | 31s | Daily activities | Homes | 58.4 |
| IndianRoad (CG-DETR) | 1231 | 26,863 | 60s | Open | Self-collected | 5.1 |

## 3.3 Video Moment Retrieval

**Dataset Structure:**  For the Video Moment Retrieval task, we annotated 26863 queries, 21,477 for training, and 5,386 for testing. Our query is like "Car is doing lane changing with clear lane markings.", "MotorBike runs in the wrong lane.", "Motorized Tricycle is overtaking.". Those queries are very challenging since some actors are not usual in most visual encoder training data. The actions require the reasoning of the actor, the nearby agents, and the environment.

**Experiment Setting:**  Following CG-DETR Moon et al. (2023) on Charades-STA, we utilize slow-fast and CLIP backbone features. The model is trained with a batch size of 32 over 200 epochs, employing a learning rate of $2 \times 10^{-4}$ without any learning rate drop. To accommodate adaptive cross-attention mechanisms, 45 dummy tokens are utilized. The selection process for moment-representative saliency involves pooling 10 candidates, from which 2 are chosen. The architecture includes 3 transformer encoder layers, 3 transformer decoder layers, and 2 layers each for adaptive cross-attention and dummy encoding. Additionally, there is 1 layer each dedicated to moment and sentence encoding. The loss function coefficients are set uniformly to 1 for most, except for highlight detection and distillation where they are increased to 4 and 10 respectively, to emphasize their importance in the training process. These settings are meticulously chosen to enhance the model's ability to understand and generate accurate moment retrievals.

**Results:**  As shown in Table 4, R1@0.5 refers to a metric that evaluates the model's ability to rank the most relevant moment within the top 1 results, with a minimum overlap of 50% between the predicted and ground-truth moment durations. The CG-DETR method only gets 5.1 R1@0.5 on IndianRoad, the perception performance degrades significantly illustrating that Video Moment Retrieval is still a challenging problem in the unstructured environment.

## 3.4 Spatiotemporal Action Localization

**Dataset Structure:**  The IndianRoad dataset stands out as a premier choice for Spatiotemporal Action Localization, thanks to its comprehensive provision of bounding box annotations and associated behavior labels, encompassing more than 2 million annotated frames. For Spatiotemporal Action Localization, we set the allocation as 1000 video clips for the training phase and 231 clips designated for the testing process. Adhering to established benchmark protocols, our evaluation encompasses 16 distinct behavior classes, employing the mean Average Precision (mAP) as the evaluation metric, predicated on a frame-level Intersection over Union (IoU) threshold set at 0.5.

**Experiment Setting:**  The spatiotemporal action localization pipeline includes detections and recognition. For the object detection, we use the Swin-T detector in Section 3.2. For recognition network, following ACAR-Net Pan et al. (2021), we conduct experiments using a SlowFast R-101, pre-trained on the Kinetics-700 dataset Carreira et al. (2019), without non-local blocks. The inputs are 64-frame clips, where we sample $T = 8$ frames with a temporal stride $\tau = 8$ for the slow pathway, and $\alpha T(\alpha = 4)$ frames for the fast pathway. We train ACAR-Net using synchronous SGD with a batch size of 16. For the first 3 epochs, we use a base learning rate of 0.008, which is then decreased by a factor of 10 at iterations 4 epochs and 5 epochs. We use a weight decay of $1 \times 10^{-7}$ and Nesterov momentum of 0.9. We use both ground-truth boxes and predicted object boxes for

training. For inference, we scale the shorter side of input frames to 384 pixels and use detected object boxes with scores greater than 0.85 for final behavior classification.

**Results:** As shown in Table 5, ACAR-Net gets 6.3% mAP on IndianRoad versus 33.3% on AVA v2.2, which shows IndianRoad is a very challenging dataset and has tremendous room to improve. IndianRoad's complexity arises from diverse agents (16 categories VS 1 category of other human-centric datasets), fast and varied motion patterns, and dense traffic. It offers valuable resources to improve multi-agent behavior recognition.

Table 5: Spatiotemporal Action Localization. ACAR-Net gets 6.3% mAP on IndianRoad, which shows IndianRoad is a very challenging dataset and has tremendous room to improve.

| Dataset | Bbox # | Instance # | Video # | Actor class | Action class | Resource | SOTA (mAP) |
|---|---|---|---|---|---|---|---|
| UCF101-24 Soomro et al. (2012) | 574k | 4458 | 3207 | - | 24 | YouTube | 90.3 |
| J-HMDB Jhuang et al. (2013) | 32k | 928 | 928 | - | 21 | Movies, YouTube | 83.8 |
| AVA v2.2 Gu et al. (2018) | 426k | 386k | 430 | 1 | 80 | Movies, YouTube | 45.1 |
| AVA v2.1 Gu et al. (2018) | 426k | 386k | 430 | 1 | 80 | Movies, YouTube | 41.7 |
| MultiSports Li et al. (2021) | 902k | 37701 | 3200 | 1 | 66 | YouTube | 8.8 |
| AVA v2.2 (ACAR) Gu et al. (2018) | 426k | 386k | 430 | 1 | 80 | Movies, YouTube | 33.3 |
| IndianRoad | 1600k | / | 1231 | 16 | 16 | self-collected | 6.3 |

Table 6: Multi-label Video Action Recognition. SlowFast achieves 4.2% more performance on Charedes than IndianRoad, which means IndianRoad is harder.

| Dataset | Size | Video # | Actions per video | Labelled instances | domain | SOTA (mAP) |
|---|---|---|---|---|---|---|
| Charades Sigurdsson et al. (2016b) | / | 9,848 | 6.8 | 67k | Daily Activities | 66.3 |
| Charades (SlowFast) Sigurdsson et al. (2016b) | / | 9,848 | 6.8 | 67k | Daily Activities | 45.2 |
| IndianRoad (SlowFast) | 1920×1080 | 10,083 | 1-13 | 1.6M | Outdoor Actions | 41.0 |

## 3.5 Multi-label Video Action Recognition

**Dataset Structure:** IndianRoad for Multi-label Video Action Recognition dataset is composed of 10,083 videos clips, involving interactions with 16 actors classes in 16 types of driving behavior action classes. Following the standard split, it has 8,166 training video and 1,917 validation video.

**Experiment Setting:** Following SlowFast Feichtenhofer et al. (2019), for the temporal domain, we randomly sample a clip from the full-length video. For the spatial domain, we randomly crop 224×224 pixels from a video, or its horizontal flip, with a shorter side randomly sampled in [256, 320] pixels. Performance is measured in mean Average Precision (mAP).

**Results:** As shown in Table 6, SlowFast Feichtenhofer et al. (2019) gets 41.0 mAP when using Kinetics-600 pre-trained model on IndianRoad. SlowFast achieves 4.2% more performance on Charedes, which means IndianRoad is harder in terms of Multi-label Video Action Recognition task.

## 4 Conclusion, Limitations, and Future Work

We present a new video dataset, IndianRoad, which provides a new benchmark for video recognition research. It is a robust platform for developing, testing, and refining algorithms capable of handling the complexity of real-world environments. Through its diverse actor categories, range of actions, and unstructured nature of its video content, IndianRoad represents a significant step forward in the quest for models that can truly understand and interpret the visual world around the ego-actor. The limitation of this dataset is that we don't have segmentation and lane marking information. And it focuses on very hard scenarios, which may be very challenging for most perception models. In the future, we would like to annotate the segmentation and lane marking information and gather more annotation information, which could allow more for fine-grained tasks.

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

## A  APPENDIX

### A.1  MORE RELATED DATASETS

#### A.1.1  TRACKING

The field of object tracking has significantly advanced with the development and introduction of various benchmark datasets, which are crucial for evaluating the performance of tracking algorithms. One of the earliest and most widely used datasets is the OTB dataset, introduced by Wu et al. Wu et al. (2013), which has played a pivotal role in benchmarking the accuracy and robustness of trackers. The OTB dataset provides comprehensive ground truth for various objects across numerous videos, allowing for a detailed analysis of tracking algorithms under different conditions. Following the OTB, the VOT Kristan et al. (2015) challenge has introduced datasets annually since 2013, with each iteration presenting new challenges and advancements over the previous versions. The VOT challenge datasets are known for their rigorous annotation protocols and have introduced several innovations in evaluation methodologies, such as the no-reset evaluation protocol and real-time tracking evaluations. Another significant contribution to the field is TrackingNet Muller et al. (2018), which provides a large-scale dataset covering a wide variety of objects and scenarios. The LaSOT dataset by Zhan et al. Zhan et al. (2019) further extends the boundaries by offering a large-scale, high-quality dataset with lengthy video sequences and is aimed at evaluating the long-term

capabilities of tracking algorithms. LaSOT provides detailed annotations and a diverse set of challenges, making it an invaluable resource for developing and testing long-term trackers. The GOT-10k dataset by Huang et al. Huang et al. (2019) introduces a unique approach by focusing on a wide variety of object classes with a zero-shot evaluation protocol. This dataset challenges trackers to perform well on previously unseen objects, pushing the boundaries of generalization in object tracking. PoseTrack Zhang et al. (2021) and GDTM Liu et al. (2024) focus more on specialized datasets. PointOdyssey Zheng et al. (2023) is a synthetic dataset specifically designed for long-term point tracking, addressing the limitation of short temporal context in existing datasets.

Compared with those datasets, IndianRoad's diverse actors allow for the evaluation of robust tracking methods capable of handling occlusions, cluttered scenes, and dynamic environments. It broadens the scope of tracking scenarios, facilitating the development of algorithms capable of operating under a wider range of real-world conditions.

### A.1.2 DETECTION

In the realm of object detection, except for Pascal VOC challenge Everingham et al. (2010) and the MS COCO dataset Lin et al. (2014a), there are some specific applications such as autonomous drivingSun et al. (2020); Chandra et al. (2023), dedicated datasets have been created to address the unique challenges of this domain. Waymo Open Dataset Sun et al. (2020) represents a significant leap forward in scale and diversity for autonomous driving datasets. It encompasses a vast array of sensor data, including high-resolution LiDAR and camera footage, across a wide range of driving conditions and scenarios. This dataset has been instrumental in pushing the boundaries of perception algorithms in terms of scalability, robustness, and accuracy. The NuScenes dataset Caesar et al. (2020) is another pivotal dataset for autonomous vehicle perception, offering a rich set of sensor modalities, including RADAR, which is less common in other datasets. NuScenes provides detailed annotations for a variety of object classes in complex urban environments, making it a valuable resource for multi-modal perception systems.

Compared with those datasets, IndianRoad has more challenges in terms of the mixture of agents, area, time of the day, traffic density, and weather conditions.

### A.1.3 SPATIOTEMPORAL ACTION LOCALIZATION

Spatiotemporal action localization is a crucial task in computer vision that involves identifying both the temporal and spatial boundaries of actions within videos. This task enables the understanding of complex video content by pinpointing where and when specific actions occur. Over the years, several datasets have been introduced to facilitate research and development in this area. Here, we review some of the key datasets that have significantly contributed to advancing spatiotemporal action localization research. UCF101-24 Soomro et al. (2012) is one of the earliest datasets tailored for spatiotemporal action localization. Derived from the UCF101 dataset, it includes 24 sports categories with temporal annotations and bounding boxes around the action instances. Despite its relatively small size, UCF101-24 has been pivotal in early methodological developments. The J-HMDB dataset Jhuang et al. (2013) is another fundamental resource that consists of 21 different action classes with 928 video clips. Each action instance is annotated with a bounding box across all frames, providing detailed spatial and temporal information. The dataset's focus on human actions makes it particularly valuable for human-centered action localization research. Furthermore, MEVA Corona et al. (2021) and VIRAT Oh et al. (2011) focus on unmanned aerial vehicles and surveillance activity detection.

More recently, the MultiSports dataset Li et al. (2021) has been introduced, focusing on multi-person and multi-action scenarios within sports videos. It contains annotations for 133 action classes across more than 20 different types of sports, with precise spatiotemporal bounding boxes for each action instance. This dataset is particularly challenging due to the dynamic nature of sports, which include frequent occlusions and interactions between athletes. Our IndianRoad dataset makes the progression from relatively simple, single-action instances in constrained environments to complex, multi-action scenarios in uncontrolled environments and challenging scenarios.

### A.1.4 VIDEO MOMENT RETRIEVAL

The task of Video Moment Retrieval (VMR) involves identifying specific moments within a video that correspond to a textual query. This area has seen significant interest due to its applications in video understanding, search, and interaction. Various datasets have been introduced to facilitate research in VMR, each with its unique characteristics and challenges. This section reviews some of the key datasets that have been influential in advancing VMR research. One of the earliest and most widely used datasets in this domain is the Charades dataset by Sigurdsson et al. Sigurdsson et al. (2016a). It consists of videos of daily activities annotated with descriptions and temporal intervals. The dataset has been instrumental in developing early VMR models due to its rich annotations and the naturalistic setting of the videos. Building on the foundations laid by Charades, the ActivityNet Captions dataset Krishna et al. (2017) offers a larger scale and diversity of activities. This dataset features dense temporal annotations with corresponding natural language descriptions, making it a staple for training and evaluating VMR systems. Another significant contribution to the field is the TVR dataset Lei et al. (2020). This dataset stands out for its focus on television show episodes, providing a mix of dialogue, action, and interaction that is more complex than daily activities. The TVR dataset is particularly noted for its challenging queries that require deep understanding of both the video content and the textual descriptions. The DiDeMo dataset Hendricks et al. (2017) offers a different approach by focusing on describing distinct moments in a video with a single sentence. Its unique structure facilitates research into more granular moment retrieval and alignment between video content and textual descriptions. These datasets have collectively contributed to yhr progress in VMR by providing diverse challenges and enabling the development of advanced models capable of understanding complex video-text relations. However, the unstructured videos in IndianRoad add more sophistication and increase the complexity of tasks that models are expected to perform.

### A.1.5 MULTI-LABEL VIDEO ACTION RECOGNITION

In the field of computer vision, multi-label video action recognition has become increasingly important for applications ranging from surveillance to content analysis and retrieval. Unlike single-label action recognition, where each video is associated with a single action, multi-label video action recognition involves identifying multiple actions that occur simultaneously or sequentially within a video.

The Charades dataset by Sigurdsson et al. Sigurdsson et al. (2016b) is the most popular and is specifically designed for multi-label video action recognition. It contains 9,848 videos with an average length of 30 seconds, annotated with 157 action labels. The dataset stands out for its focus on everyday activities, with videos featuring multiple actions performed by the actors. Charades facilitates the development and evaluation of models capable of recognizing multiple simultaneous actions, making it a cornerstone in multi-label video action recognition research.

Given that Charades focuses on daily activities, it primarily includes indoor scenarios. This focus may limit the applicability of derived models for outdoor activities or other contexts not covered by the dataset. Our IndianRoad dataset makes up for the indoor limitation and introduces more complex actions, leading to the advancement of more sophisticated and accurate recognition models.

