# OpenReview forum: "IndianRoad: A Video Dataset of Diverse Atomic Visual Elements in Dense and Unpredictable Environments"
_ICLR.cc/2025/Conference — ICLR 2025 Conference Withdrawn Submission_

### Official Review · Reviewer_TwUy · 2024-11-01

**Soundness:** 2
**Presentation:** 2
**Contribution:** 1
**Rating:** 5
**Confidence:** 4

**Summary:**

This dataset is for road scenes from an ego-vehicle view in India, providing 1,231 videos, each one minute long. Annotations are included for five common tasks: 2D bounding box-based tracking and detection, spatiotemporal action localization and video moment retrieval by text query, and video action recognition involving multiple objects. Baseline models that perform reasonably well on existing comparable datasets perform poorly on this proposed dataset, demonstrating its challenging nature.

**Strengths:**

- The dataset provides challenging scenarios for tasks in existing datasets.
- Annotations are provided for five different tasks, allowing flexibility for various applications.

**Weaknesses:**

## Major
- Weak novelty;
  - previous datasets also targeting unstructured traffic scenes in India, such as those in [1,2], are not referred to. What is the novelty of this dataset compared to the existing datasets or a combination of them?
  - The manuscript does not provide new insights regarding data collection methods.
  - The annotated tasks are existing ones, and the fact that unstructured scenes are difficult is not surprising [1,2].
- No explanation of the annotation protocol is provided for quality assurance of the dataset, such as how consistency of text annotation across videos by different annotators is ensured.

## Minor
- The paper structure is hard to follow; the introduction and related work should be separated.
- The dataset’s domain is very limited, specifically to India. It would be more interesting if data were collected across different countries or regions for a more generalizable dataset.

[1] Varma et al. IDD: A Dataset for Exploring Problems of Autonomous Navigation in Unconstrained Environments. WACV, 2019.
[2] Paranjape et al. DATS_2022: A versatile Indian dataset for object detection in unstructured traffic conditions. Data in Brief, 2022.

**Questions:**

The questions regarding the unclear novelty and the annotation protocol are included in the weakness section.

---

### Official Review · Reviewer_UcpB · 2024-11-03

**Soundness:** 2
**Presentation:** 2
**Contribution:** 1
**Rating:** 3
**Confidence:** 5

**Summary:**

The authors introduce IndianRoad, a novel dataset created to evaluate perception methods with a strong emphasis on vulnerable road users (VRUs)—including pedestrians, animals, motorbikes, and bicycles—in complex, unpredictable environments. This dataset comprises 16 diverse actor categories (e.g., animals, humans, vehicles) and includes 16 distinct action types, covering complex and rare scenarios. IndianRoad features dense annotations, with over 13 million bounding boxes identifying actors, of which more than 1.6 million boxes are further annotated with detailed actor identification and action/behavior information. The authors propose tasks such as Tracking and Detection, Spatiotemporal Action Localization, Video Moment Retrieval, and Multi-label Video Action Recognition, benchmarking various video understanding baselines.

**Strengths:**

1. It’s fascinating to see the authors address the challenges faced by vulnerable road users and exciting to witness their efforts in creating a dataset that supports a wide range of tasks.
2. Labeling a large-scale dataset is a substantial and nontrivial effort. It’s commendable that the authors are willing to share their results with the community.
3. The authors conduct an extensive set of experiments, benchmarking algorithms for Tracking and Detection, Spatiotemporal Action Localization, Video Moment Retrieval, and Multi-label Video Action Recognition.

**Weaknesses:**

While it is exciting to see the efforts toward traffic scene understanding for vulnerable road users, I have the following concerns for this work.

1. **Lack of comprehensive comparison with existing traffic scene datasets:** The authors begin their introduction by discussing advancements in video understanding within the computer vision community, highlighting several recent datasets. While they briefly mention the Waymo dataset in line 149, the community has made significant efforts toward traffic scene understanding through video-based datasets. It is concerning that these contributions are overlooked. Additionally, works such as that of Chandra et al. [4], which specifically address traffic scene understanding for vulnerable road users, are not compared, leaving the unique contributions of *IndianRoad* unclarified. Please compare the number and types of annotations, the diversity of scenarios, or the representation of vulnerable road users. Additionally, please provide a detailed comparison table or discussion that highlights how IndianRoad differs from or improves upon these key datasets, particularly in terms of its focus on vulnerable road users and complex environments.

    Please find the following reference.

    Datasets:

    1. V. Ramanishka, et al., "Toward Driving Scene Understanding: A Dataset for Learning Driver Behavior and Causal Reasoning, CVPR 2018.
    2. Srikanth Malla, et al., TITAN: Future Forecast using Action Priors, CVPR 2020.
    3. Jianwu Fang, et al., DADA: Driver Attention Prediction in Driving Accident Scenarios, T-PAMI 2022.
    4. Rohan Chandra et al., METEOR: A Dense, Heterogeneous, and Unstructured Traffic Dataset With Rare Behaviors, IROS 2022.
    5. Gurkirt Singh et al., ROAD: The ROad event Awareness Dataset for Autonomous Driving, T-PAMI 2023.
    6. Yu Yao et al., When, Where, and What? A New Dataset for Anomaly Detection in Driving Videos, T-PAMI 2023.
    7. Srikanth Malla, et al., DRAMA: Joint Risk Localization and Captioning in Driving, WACV 2023.
    8. Nakul Agarwal and Yi-Ting Chen, Ordered Atomic Activity for Fine-grained Interactive Traffic Scenario Understanding, ICCV 2023.
    9. Enna Sachdeva et al., Rank2Tell: A Multimodal Driving Dataset for Joint Importance Ranking and Reasoning, WACV 2024.

2. **Limited consideration of existing traffic scene understanding algorithms in proposed benchmarks:** Similar to the first concern, the authors do not incorporate comparisons with prior studies on traffic scene understanding algorithms. Below is a list of relevant works that should be considered. Please provide justification by drawing comparisons with these studies. Please specify the lack of experiments, e.g., the missing comparisons using Action-slot [4] for multilabel action recognition and Khan et al for spatial-temporal action localization on IndianRoad. Please include these methods and compare the performance of them with benchmarked algorithms on IndianRoad. If you cannot do so, please explain why such comparisons may not be directly applicable. This would help clarify the unique challenges posed by IndianRoad.

    1. Li et al., Learning 3D-aware Egocentric Spatial-Temporal Interaction via Graph Convolutional Networks, ICRA 2020
    2. Khan et al., Spatiotemporal Deformable Scene Graphs for Complex Activity Detection, BMVC 2021
    3. Malla et al., DRAMA: Joint Risk Localization and Captioning in Driving, WACV 2023
    4. Kung et al., Action-Slot: Visual Action-centric Representation for Atomic Activity Recognition in Traffic Scenes, CVPR 2024
    5.  Khan et al., A Hybrid Graph Network for Complex Activity Detection in Video, WACV 2024

**Questions:**

1. Please clarify the unique contributions of the proposed dataset in comparison to existing traffic scene datasets.
2. Please provide justification for the absence of baseline comparisons in the experiments.

---

### Official Review · Reviewer_tPeh · 2024-11-03

**Soundness:** 2
**Presentation:** 2
**Contribution:** 3
**Rating:** 3
**Confidence:** 4

**Summary:**

The paper introduces IndianRoad, a video dataset designed to support a range of video recognition tasks in dense and unstructured road environments, particularly focusing on Indian traffic scenarios. The dataset addresses tracking, detection, spatiotemporal action localization, video moment retrieval, and multi-label video action recognition.
I am very grateful for the huge amount of work the author put into building the dataset, but there are still many problems in this article that need to be further addressed.

**Strengths:**

1. The dataset offers detailed and high-quality annotations across various weather conditions, traffic densities, and times of day.
2. By focusing on a highly variable and realistic road environment, IndianRoad enables the development of models that are better suited for real-world traffic scenarios.

**Weaknesses:**

I appreciate the amount of work the author put in, but I think the entire article reads very much like a technical report rather than an academic paper, and the writing needs a lot of revision.

1. Taking the tracking task as an example, some of the author's basic concepts are wrong. For example, the author mentioned in L651 that GOT-10k is a zero-shot evaluation, but the original text of GOT-10k says "zero-overlapped evaluation." Please note that the biggest contribution of GOT-10k is that the categories of the training set and the test set do not overlap, which can effectively measure the algorithm's generalization. Therefore, "zero-overlapped evaluation" is another form of open-set evaluation. Here, the author can use zero-overlapped evaluation or open-set evaluation to characterize the evaluation paradigm of the SOT task, but it is definitely not a zero-shot evaluation. Because the SOT task is a one-shot evaluation (only the bounding box information of the target in the first frame of each sequence can be used), it is not a zero-shot. This is a problem with the essential definition of the task, but unfortunately, the author has not fully understood the essential characteristics of SOT.

2. I am confused because the dataset examples the author gave are more inclined to MOT, but both the datasets used for comparison and the tracking algorithms and indicators used are SOT. Both SOT and MOT are tracking tasks, but they are two completely different directions. The author mixed these two tasks together, which made me very confused.

3. There are many irregularities in the writing. For example, we need to add commas to numbers as a standard expression. However, the author added commas in some places and not in others, and even Table 4 has both added and not added forms. In addition, the form of the table is not a standard three-line table, and it even uses a variety of table drawing methods, which makes it look very irregular.

4. I think there are some problems with the author's related work. For example, the author completely mistyped the author's information for the LaSOT dataset. I am unsure whether the author has carefully read the original article when researching related work. In addition, when introducing the datasets related to the tracking task, the author tried to emphasize the diversity and complexity of this work. However, in recent years, SOT datasets such as VideoCube (Global Instance Tracking: Locating Target More Like Humans, TPAMI 2023) have achieved innovation in data scale and scene complexity, and the VastTrack (VastTrack: Vast Category Visual Object Tracking, NeurIPS 2024) dataset also far exceeds other tracking datasets in data volume and complexity. The author did not introduce or discuss these high-quality related works.

To sum up, I think what the author needs to consider is not a comprehensive but general introduction of what tasks his dataset can support, which will cause readers to miss the point completely; instead, he should fully understand the characteristics of other work and find the biggest difference between his own work, and then use this as the core for discussion, so that readers can more clearly understand the author's motivation and ideas for building the dataset.

**Questions:**

Please see the weakness

---

### Official Review · Reviewer_DJTC · 2024-11-04

**Soundness:** 3
**Presentation:** 3
**Contribution:** 4
**Rating:** 5
**Confidence:** 4

**Summary:**

This paper focused on introducing a dataset called IndianRoad. It features videos captured in diverse real-world settings, encompassing various weather conditions, times of day, road scenarios, and traffic densities. One major motivation is to provide a benchmark dedicated to Asian traffic scenarios, e.g., Indian traffic in this paper, which are far more complex, involving numerous objects with distinct motions and behaviors.

IndianRoad serves as a comprehensive benchmark for various video tasks, including tracking, detection, spatiotemporal action localization, language-visual moment retrieval, and multi-label video action recognition.

**Strengths:**

IndianRoad features 16 diverse actor categories—including animals, humans, and various vehicle types—and 16 complex action types, such as cut-ins, zigzag movements, and U-turns, which demand advanced reasoning capabilities. The dataset includes over 13 million densely annotated bounding boxes with actor identification, of which more than 1.6 million also include detailed annotations of actions and behaviors. The dataset holds values for evaluating perception methods, particularly due to its high representation of Vulnerable Road Users.

**Weaknesses:**

An essential experiment that has yet to be conducted across all tasks is to demonstrate the value of integrating the IndianRoad dataset with other existing datasets as training data. This analysis should evaluate whether this combination results in a performance boost compared to relying solely on the existing datasets without incorporating the newly introduced dataset.

Claiming that IndianRoad is a more challenging dataset may be misleading, as the authors used pre-trained models from existing datasets to evaluate it. This conclusion could be problematic, as the evaluation inevitably introduces a data distribution shift due to the differences between the existing datasets and IndianRoad. The authors need to address this potential issue to provide a clearer assessment of the dataset's challenges. For instance, the authors are encouraged to use a portion of the IndianRoad dataset as the training set and train the models to evaluate their performance on the test split of the IndianRoad dataset. This approach will help determine whether the IndianRoad dataset presents the anticipated challenges.

**Questions:**

The criteria and definitions for the state-of-the-art (SOTA) approaches referenced in each task are unclear. Are the authors referring to recent advancements in SOTA research, or are they citing classical methods as SOTAs? The reviewer observed that most of the SOTA approaches cited are from 2018 and 2019. It is recommended that the authors clarify their definition of SOTA and consider including more recent methods to provide a comprehensive overview.

**Details Of Ethics Concerns:**

The paper claimed "To protect privacy, we will hide identities by blurring the faces of persons and license plates of vehicles
in the dataset with blurring techniques (face detection method Retinaface Deng et al. (2020), license
plates method Yan et al. (2023)) to ensure that the identity of pedestrians and other individuals (cars)
is not discernible." However, the released demo video prominently displays numerous license plates, raising concerns about potential identity leakage for captured vehicles and pedestrians on Indian roads.

---

### Note · Authors · 2024-11-13

I have read and agree with the venue's withdrawal policy on behalf of myself and my co-authors.